# A Deep Learning Approach for Survival Clustering without End-of-life Signals

## Abstract

The goal of survival clustering is to map subjects (e.g., users in a social network, patients in a medical study) to $K$ clusters ranging from low-risk to high-risk. Existing survival methods assume the presence of clear *end-of-life* signals or introduce them artificially using a pre-defined timeout. In this paper, we forego this assumption and introduce a loss function that differentiates between the empirical lifetime distributions of the clusters using a modified Kuiper statistic. We learn a deep neural network by optimizing this loss, that performs a soft clustering of users into survival groups. We apply our method to a social network dataset with over 1M subjects, and show significant improvement in C-index compared to alternatives.

## 1 Introduction

Free online subscription services (e.g., Facebook, Pandora) use survival models to predict the relationship between observed subscriber covariates (e.g. usage patterns, session duration, gender, location, etc.) and how long a subscriber remains with an active account (Kapoor et al., 2014; Ciampaglia and Taraborelli, 2015). Using the same tools, healthcare providers make extensive use of survival models to predict the relationship between patient covariates (e.g. smoking, administering drug A or B) and the duration of a disease (e.g., herpes, cancer, etc.). In these scenarios, rarely there is an end-of-life signal: non-paying subscribers do not cancel their accounts, tests rarely declare a patient cancer-free. We want to assign subjects into $K$ clusters, ranging from short-lived to long-lived subscribers (diseases).

Despite the recent community interest in survival models (Alaa and van der Schaar, 2017; Luck et al., 2017), existing survival analysis approaches require an unmistakable end-of-life signal (e.g., the subscriber deletes his or her account, the patient is declared disease-free), or a pre-defined end-of-life "timeout" (e.g., the patient is declared disease-free after 5 years, the subscriber is declared permanently inactive after 100 days of inactivity). Methods that require end-of-life signals also include (Iorio et al., 2009; Bohlourihajjar and Khazaei, 2017; Bair and Tibshirani, 2004; Eleuteri et al., 2003; 2007; Ishwaran et al., 2010; Lagani and Tsamardinos, 2010; LeBlanc and Crowley, 1993; Witten and Tibshirani, 2010b; Bøvelstad et al., 2007; Hothorn et al., 2006; Shivaswamy, Chu, and Jansche, 2007; Shipp et al., 2002; Gaynor and Bair, 2013; Yang et al., 2010; Kapoor et al., 2014; Aggarwal, Gates, and Yu, 2004; Basu, Banerjee, and Mooney, 2002; Basu, Bilenko, and Mooney, 2004; Nigam et al., 1998; Witten and Tibshirani, 2010a; Law, Urtasun, and Zemel, 2017).

In this work, we propose to address the lifetime clustering problem without end-of-life signals for the first time, to the best of our knowledge. We begin by describing two possible datasets where such a clustering approach could be applied.

- Social Network Dataset : Users join the social network at different times and participate in activities defined by the social network (login, send/receive comments). The covariates are the various attributes of a user like age, gender, number of friends, etc., and the inter-event time is the time between user's two consecutive activities. In this case, censoring is due to a fixed point of data collection that we denote $t_m$, the time of measurement. Thus, time till censoring for a particular user is the time from her last activity to $t_m$. Lifetime of a user is defined as the time from her joining till she permanently deletes her account.

- Medical Dataset : Subjects join the medical study at the same time and are checked for the presence of a particular disease. The covariates are the attributes of the disease-causing cell in subject, inter-event time is the time between two consecutive observations of the presence of disease. The time to censoring is the difference between the time of last observation when the disease was present and the time of final observation. If the final observation for a subject indicates presence of the disease, then time to censoring is zero. Lifetime of the disease is defined as the time between the first observation of the disease and the time until it is permanently cured.

We use a deep neural network and a new loss function, with a corresponding backpropagation modification, for clustering subjects without end-of-life signals. We are able to overcome the technical challenges of this problem, in part, thanks to the ability of deep neural networks to generalize while overfitting the training data (Zhang et al., 2017). The task is challenging for the following reasons:

- The problem is fully unsupervised, as there is no pre-defined end-of-life timeout. While semi-supervised clustering approaches exist (Aggarwal, Gates, and Yu, 2004; Basu, Banerjee, and Mooney, 2002; Basu, Bilenko, and Mooney, 2004; Nigam et al., 1998; Witten and Tibshirani, 2010a), they assume that end-of-life signals appearing before the observation time are observed; to the best of our knowledge, there are no fully unsupervised approach that can take complex input variables.
- There is no hazard function that can be used to define the "cure" rate, as we cannot determine whether the disease is cured, or whether the subscriber will never return to the website, without observing for an infinitely long time.
- Cluster assignments may depend on highly complex interactions between the observed covariates and the observed events. The unobserved lifetime distributions may not be smooth functions.

**Contributions.** Using the ability of deep neural networks to model complex nonlinear relationships in the input data, our contribution is a loss function (using the p-value from a modified Kuiper nonparametric two-sample test (Kuiper, 1960)) and a backpropagation algorithm that can perform model-free (nonparametric) unsupervised clustering of subjects based on their latent lifetime distributions, even in the absence of *end-of-life* signals. The output of our algorithm is a trained deep neural network classifier that can (soft) assign test and training data subjects into $K$ categories, from high-risk and to low-risk individuals. We apply our method to a large social network dataset and show that our approach is more robust than competing methods and obtains better clusters (higher C-index scores).

**Why deep neural networks.** As with any optimization method that returns a point estimate (a set of neural network weights $W$ in our case), our approach is subject to overfitting the training data. And because our loss function uses $p$-values, the optimization and overfitting have a rather negative name: $p$-hacking (Nuzzo, 2014). That is, the optimization is looking for a $W$ (*hypothesis*) that decreases the $p$-value. Deep neural networks, however, are known to both overfit the training data and generalize well Zhang et al. (2017). That is, the hypothesis ($W$) tends to also have small $p$-values in the (unseen) test data, despite overfitting in the training data ($p$-hacking).

**Outline**: In section 3, we describe the traditional survival analysis concepts that assume the presence of *end-of-life* signals. In section 4, we define a loss function that quantifies the divergence between empirical lifetime distributions of two clusters without assuming *end-of-life* signals. We also provide a neural network approach to optimize said loss function. We describe the dataset used in our experiments followed by results in section 5. In section 6, we describe a few methods in literature that are related to our work. Finally, we present our conclusions in section 7.

## 2 FORMAL FRAMEWORK

In this section, we formally define the statistical framework underlying the clustering approach introduced later in this paper. We begin by defining the datasets relevant to the survival clustering task.

**Definition 1** (Dataset). *Dataset $\mathcal{D}$ consists of a set of $n$ subjects with each subject $u$ having the following observable quantities $\Psi_u = \{X_u, \{Y_{u,i}\}_{i=1}^{q_u}, S_u\}$, where $X_u$ are the covariates of subject*

$u$, $\{Y_{u,i}\}_{i=1}^{q_u}$ *are the observed inter-event times (disease outbreaks, website usage), $q_u$ is the number of observed events of $u$, and $S_u$ is the time till censoring.*

Note that the two example datasets described in section 1 fit into this definition. For instance, in the social network dataset, for a particular user $u$, $X_u$ is a vector of her covariates (such as age, gender, etc.), $Y_{u,i}$ is the time between her $i$th and $(i-1)$st activity (login, send/receive comments), and her time till censoring is given by, $S_u = t_m - \sum_i Y_{u,i}$, where $t_m$ is the time of measurement.

Next, we define the lifetime clustering problem applicable to the aforementioned datasets.

**Definition 2** (Clustering problem). *Consider a dataset of $n$ subjects, $\mathcal{D}$, constructed as in definition 1. Let $\hat{P}(U_k)$ be the* latent *lifetime distribution of all subjects $U_k = \{u\}$ that belong to cluster $k \in \{1, \ldots, K\}$. Our goal is to find a mapping $\kappa : X_u \to \{1, \ldots, K\}$, of covariates into clusters, in the set of all possible such mappings $\mathcal{K}$, that maximizes the divergence $d$ between the* latent *lifetime distributions of all subjects:*

$$\kappa^\star = \arg\max_{\kappa \in \mathcal{K}} \sum_{i=1}^{K} \sum_{j=1}^{K} \mathbf{1}\{i \neq j\} d(\hat{P}(U_i(\kappa)), \hat{P}(U_j(\kappa))), \tag{1}$$

*where $U_k(\kappa)$ is the set of users in $U$ mapped to cluster $k$ through $\kappa$, and $d$ is a distribution divergence metric. $\kappa^*$ optimized in this fashion clusters the subjects into low-risk/high-risk groups.*

However, because $\hat{P}(U_k)$ are latent distributions, we cannot directly optimize Eq.(1). Rather, our loss function must provide an indirect way to optimize Eq.(1) without end-of-life signals. In what follows, we define the activity process of subjects in cluster $k$ as a Random Marked Point Processes (RMPP).

**Definition 3** (Observable RMPP cluster process). *Consider the $k$-th cluster. The RMPP is $\Phi_k = \{X_k, \{(A_{k,i}, Y_{k,i})\}_{i \in \mathbb{Z}^*}, S_k\}$, where $Y_{k,i}$ is the inter-event time between the $(i-1)$-st and the $i$-th activities, $X_k$ are the random variable representing the covariates of subjects in cluster $k$, $S_k$ is the time from last event until censoring at cluster $k$, and $A_{k,i} = 0$ indicates an activity with an end-of-life signal, otherwise $A_{k,i} = 1$. All these variables may be arbitrarily dependent. This definition is model-free, i.e., we will not prescribe a model for $\Phi_k$.*

Note that, at least theoretically, $\Phi_k$ continues to evolve beyond the end-of-life signal, but this evolution will be ignored as it is irrelevant to us. The relative time of the $i$-th activity of a subject of cluster $k$, since the subject's first activity, is $\sum_{i' \leq i} Y_{k,i'}$, as long as we haven't seen an end-of-life signal, i.e., $\prod_{i' < i} A_{k,i} = 1$.

**Definition 4** (RMPP Lifetime). *The random variable that defines the lifetime of a subject of cluster $k$ is*

$$T_k := \max_i \left( \sum_{i' \leq i} Y_{k,i'} \prod_{i'' < i} A_{k,i''} \right). \tag{2}$$

We now define censored lifetimes using $\Phi_k$.

**Definition 5** (RMPP Censored Lifetimes). *The random variable that defines the last observed action time of a subject $u$ of cluster $k$ is*

$$H_k := \max_i \left( \delta_{k,i} \sum_{i' \leq i} Y_{k,i'} \prod_{i' < i} A_{k,i} \right). \tag{3}$$

*where $\delta_{k,i} = \mathbf{1}\{\sum_{i' \leq i} Y_{k,i'} \leq S_k\}$.*

Let $i^\star(S_k)$ be a random variable that denotes the number of events until the censoring time $S_k$. The main challenge is not knowing when $H_k = T_k$, because we are unable to observed the end-of-life signal $A_{k,i^\star(S_k)} = 0$. Clearly, this affects the decision of which subjects have been censored and which have not. Later, we introduce probability of *end-of-life*, $p : (X_u, S_u) \to [0, 1]$, that provides a way around this challenge.

## 3 BACKGROUND

In this section, we review the major concepts in survival analysis that are used in this paper. Let $T_u$ denote the lifetime of a subject $u$. For now, our description assumes an **Oracle** that provides *end-of-life* signals. Thus, in addition to $\Psi_u$, we assume for each subject $u$ and, another observable quantity, $A_{u,i}$ that denotes whether *end-of-life* has been reached at the user's $i$th activity. In survival applications, $A_u$ is typically used to specify if the required event did not occur until the end of study, known as *right-censoring*. We shall forego this assumption in subsequent sections and provide a way around the lack of these signals.

**Lifetime distribution & Hazard function (Oracle).** Lifetime (or survival) distribution is defined as the probability that a subject $u$ survives at least until time $t$,

$$S_u(t) = P[T_u > t] = 1 - F_u(t), \qquad 0 < t < \infty, \tag{4}$$

where $F_u(t)$ is the cumulative distribution function of $T_u$.

In survival applications, it is typically convenient to define the hazard function, that represents the instantaneous rate of death of a subject given that she has survived until time $t$. The hazard function of a subject $u$ is $\lambda_u(t) = \frac{dF_u(t)}{S_u(t)}$, where $dF_u$ is the probability density of $F_u$.

**Kaplan-Meier Estimates and the Cox Model of Lifetime Distribution (Oracle).** Due to right-censoring, we do not observe the true lifetimes of the subjects even in the presence of *end-of-life* signals, $A_u$. We define the *observed lifetime* of subject $u$, $H_u$, as the difference between the time of first event and time of last observed event, i.e.,

$$H_u = \sum_{i=1}^{q_u} Y_{u,i} - Y_{u,1} . \tag{5}$$

Kaplan and Meier (1958) provide a way to estimate the lifetime distribution for a set of subjects while incorporating the right censoring effect. The Kaplan-Meier estimates of lifetime distribution are given by,

$$S(t; \mathcal{D}) = \prod_{\forall j \leq t} \theta_j = \prod_{\forall j \leq t} \frac{r_j - d_j}{r_j} , \tag{6}$$

where $d_j = \sum_{u \in \mathcal{D}} \mathbb{I}[H_u = j] \cdot (1 - A_{u,q_u})$ denotes the number of subjects with *end-of-life* at time $j$, and $r_j = \sum_{u \in \mathcal{D}} \mathbb{I}[H_u \geq j]$ denotes the number of subjects at risk just prior to time $j$.

Cox regression model (Cox, 1992) is a widely used method in survival analysis to estimate the hazard function $\lambda_u(t)$ using the covariates, $X_u$, of a subject $u$. The hazard function has the form, $\lambda(t|X_u) = \lambda_0(t) \cdot e^{\{\beta^T X_u\}}$, where $\lambda_0(t)$ is a base hazard function common for all subjects, and $\beta$ are the regression coefficients. The model assumes that the ratio of hazard functions of any two subjects is constant over time. This assumption is violated frequently in real-world datasets (Li et al., 2015). A near-extreme case when this assumption does not hold is shown in Figure 1(c), where the survival curves of two groups of subjects cross each other.

**Survival Based Clustering Methods (Oracle).** There have been relatively fewer works that perform survival based clustering. Bair and Tibshirani (2004) proposed a semi-supervised method for clustering survival data in which they assign Cox scores (Cox, 1992) for each feature in their dataset and considered only the features with scores above a predetermined threshold. Then, an unsupervised clustering algorithm, like k-means, is used to group the individuals using only the selected features.

Gaynor and Bair (2013) proposed supervised sparse clustering as a modification to the sparse clustering algorithm of Witten and Tibshirani (2010a). The sparse clustering algorithm has a modified k-means score that uses distinct weights in the feature set. Supervised sparse clustering initializes these feature weights using Cox scores (Cox, 1992) and optimizes the same objective function.

Both these methods assume the presence of *end-of-life* signals. In this paper, we consider the case when *end-of-life* signals are not available. We provide a loss function that quantifies the divergence between survival distributions of the clusters, and we minimize said loss function using a neural network in order to obtain the optimal clusters.

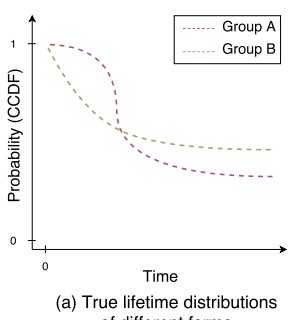 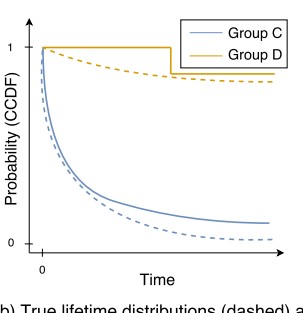 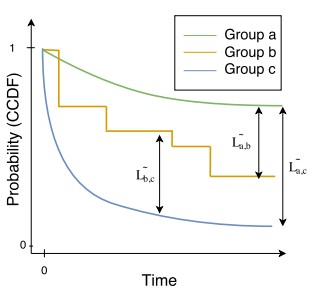

(a) True lifetime distributions of different forms

(b) True lifetime distributions (dashed) and empirical lifetime distributions (solid) of groups using KL divergence

(c) Empirical distributions of three groups showing that the Kuiper loss is not a metric

Figure 1: (a) Lifetime distributions can have different shapes (thus, a nonparametric approach). Also, distributions can cross each other thus violating the proportional hazards assumption (drawback with Logrank test) (b) KL divergence does not account for the uncertainty in the distributions and can lead to highly imbalanced groups (c) $\tilde{\mathcal{L}}$ (= $-\mathcal{L}$) is not a metric. Due to very few samples in group $b$, $\tilde{\mathcal{L}}_{a,b}$ and $\tilde{\mathcal{L}}_{b,c}$ are very low compared to $\tilde{\mathcal{L}}_{a,c}$ such that $\tilde{\mathcal{L}}_{a,c} > \tilde{\mathcal{L}}_{a,b} + \tilde{\mathcal{L}}_{b,c}$, hence violating triangle inequality.

## 4 METHODOLOGY

Our goal is to cluster the subjects into $K$ clusters ranging from low-risk to high-risk by keeping the empirical lifetime distributions of these $K$ groups as different as they can be, while ensuring that the observed difference is statistically significant. In this section, we assume there are *no end-of-life* signals.

### 4.1 LOSS FUNCTION

We introduce a loss function that is based on a divergence measure between empirical lifetime distributions of two groups, and at the same time takes into account the uncertainty regarding the *end-of-life* of the subjects. Instead of a clear *end-of-life* signal, we specify a probability for each subject $u$ that represents how likely her last observed activity coincides with her *end-of-life*.

**Definition 6** (Probability of *end-of-life*). *Given a dataset $\mathcal{D}$ (Definition 1), we define a function, by an abuse of notation, $p(X_u, S_u) \rightarrow [0, 1]$ that gives a probability of end-of-life of each subject $u$.*

Divergence measures like Kullback-Leibler divergence and Earth-Mover's distance that do not incorporate the empirical nature of the given probability distributions are not appropriate for our task as they do not discourage highly imbalanced groups (Figure 1b). This motivates the use of two-sample tests that allow for the probability distributions to be empirical. Logrank test (Mantel, 1966; Peto and Peto, 1972; Bland and Altman, 2004) is commonly used to compare groups of subjects based on their hazard rates. However, the test assumes proportional hazards (section 3) and will not be able to find groups whose hazard rates are not proportional to each other (Figure 1a). Fleming et al. (1980) introduced Modified Kolmogorov-Smirnov (MKS) statistic that works for arbitrarily right-censored data and does not assume hazards proportionality. But MKS suffers from the same drawback as the standard Kolmogorov-Smirnov statistic, namely that it is not sensitive to the differences in the tails of the distributions. In this paper, we use p-value from the Kuiper statistic (Kuiper, 1960) which extends the Kolmogorov-Smirnov statistic to increase the statistical power of distinguishing distribution tails (Tygert, 2010).

**Definition 7** (Optimization of Kuiper loss). *Given a dataset $\mathcal{D}$ (Definition 1), we define a loss $\mathcal{L}(\kappa, p)$ where, by an abuse of notation, $\kappa(X_u) \rightarrow [0, 1]$ is a mapping that performs soft clustering of subjects into two clusters 0 & 1 by outputting a probability of a subject belonging in cluster 0, and $p(X_u, S_u) \rightarrow [0, 1]$ is a function that gives a probability of end-of-life of a subject in $\mathcal{D}$. Our goal is to obtain*

$$\hat{\kappa}, \hat{p} = \arg\min_{\kappa, p} \mathcal{L}(\kappa, p),$$

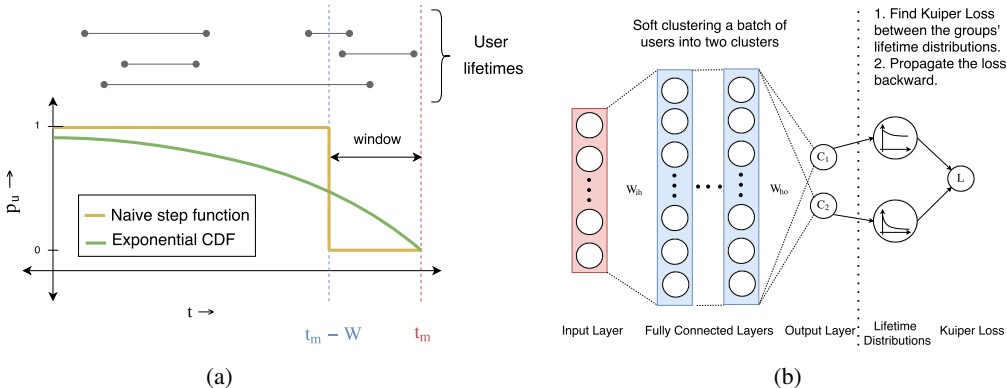

Figure 2: (a) *End-of-life* probability defined by non-decreasing functions of the difference between the time of last observed activity and the time of measurement. Figure depicts two such functions. (b) A feedforward neural network that clusters the user by optimizing the Kuiper loss.

*where the loss function*

$$\mathcal{L}(\kappa, p) = \log \left\{ 2 \sum_{j=1}^{\infty} (4j^2 \left[\lambda(\kappa, p)\right]^2 - 1) \mathrm{e}^{-2j^2 [\lambda(\kappa, p)]^2} \right\} , \qquad (7)$$

*returns the logarithm of a $p$-value from the Kuiper statistic (Press et al., 1996), with*

$$\lambda(\kappa, p) = \left( \sqrt{n(\kappa)} + 0.155 + \frac{0.24}{\sqrt{n(\kappa)}} \right) \cdot \left( D^+(\kappa, p) + D^-(\kappa, p) \right),$$

*and* $D^+(\kappa, p) = \sup_t \{S_0(t; \kappa, p) - S_1(t; \kappa, p)\}$, $D^-(\kappa, p) = \sup_t \{S_1(t; \kappa, p) - S_0(t; \kappa, p)\}$, $n(\kappa) = \frac{n_0(\kappa) \cdot n_1(\kappa)}{n_0(\kappa) + n_1(\kappa)}$, *and for* $k = 0, 1$,

$$S_k(t; \kappa, p) = \prod_{\forall j \leq t} \frac{r_j(\kappa) - d_j(\kappa, p)}{r_j(\kappa)}, \qquad (8)$$

*where* $d_j(\kappa, p) = \sum_{u \in \mathcal{D}} \mathbb{I}[H_u = j] \cdot p(X_u, S_u) \cdot \alpha_{u,k}(\kappa)$, $r_j(\kappa) = \sum_{u \in \mathcal{D}} \mathbb{I}[H_u \geq j] \cdot \alpha_{u,k}(\kappa)$, *with* $H_u$ *computed from* $\{Y_{u,i}\}_{i=1}^{q_u}$ *in eq. (5), and* $n_k(\kappa) = \sum_u \alpha_{u,k}(\kappa)$, $\alpha_{u,k}(\kappa) = \kappa(X_u)^{1-k} \cdot (1 - \kappa(X_u))^k$.

The following theorem states a few properties of our loss function.

**Theorem 1** (Kuiper loss properties). *From Definition 7, consider two clusters with true lifetime distributions $\hat{P}(U_1)$ and $\hat{P}(U_2)$. Assume an infinite number of samples/subjects. Then, the loss function defined in equation* (7) *has the following properties:*

  (a) *If the two clusters have distinct lifetime distributions, i.e. $\hat{P}(U_1) \neq \hat{P}(U_2)$ then, either $\exists \hat{\kappa}, \hat{p}$ such that $\mathcal{L}(\hat{\kappa}, \hat{p}) \to -\infty$, or $\forall \kappa, p, \mathcal{L}(\kappa, p) \to 0$.*

  (b) *If the two clusters have the same stochastic process $\Psi_u$ (Definition 1), $\Psi_u = \Psi_v$, for any two subjects $u$ and $v$, regardless of cluster assignments, then $\forall \kappa, p, \mathcal{L}(\kappa, p) \to 0$.*

We prove Theorem 1 in Appendix 8.1 by defining the activity process of the subjects using shifted Random Marked Point Processes. The loss defined above solves all the aforementioned issues; a) does not need clear *end-of-life* signals, b) use of a p-value forces sufficient number of examples in both groups, c) does not assume proportionality of hazards and works even for crossing survival curves, and d) accounts for differences at the tails.

## 4.2 NEURAL NETWORK APPROACH TO OPTIMIZE KUIPER LOSS

In this section, we describe the functions $\kappa(\cdot)$ and $p(\cdot)$ in definition 7 $\kappa(\cdot)$ gives the probability of a subject $u$ being in cluster 0, and we define it using a neural network as follows,

$$\kappa(X_u) := \sigma \left( b_L + W_L \cdot \phi(\ldots \phi(b_2 + W_2 \cdot \phi(b_1 + W_1 \cdot X_u)) \ldots) \right) , \qquad (9)$$

where $\{W_i, b_i\}_{i=1}^L$ are the weights and the biases of a neural network with $L-1$ hidden layers, $X_u$ are the covariates of subject $u$, $\phi$ is an activation function (tanh or relU in our experiments), and $\sigma$ is the softmax function. An example of a feedforward neural network that optimizes Kuiper loss is shown in figure 2b.

Next, we describe the form of *end-of-life* probability function, $p(\cdot)$. We make the reasonable assumption that $p(\cdot)$ is an increasing function of $S_u$. For example, consider two subjects $a$ and $b$, with last activities one year and one week before their respective time of censoring. Clearly, it is more likely that subject $a$'s activity is her last one than that $b$'s activity is her last one. In our experiments, we also assume that $p(\cdot)$ only depends on $S_u$, and not on the covariates $X_u$. Survival tasks commonly use the following naive technique to identify the *end-of-life* signal. They define $p(\cdot)$ using a step function, $p(X_u, S_u) = \mathbf{1}[S_u > W]$, where $W$ is the size of an arbitrary window from the time of censoring (see Figure 2a). However, this approach does not allow learning of the window size parameter $W$, and hence, the analysis can be highly coupled with the choice of $W$.

We remedy this by choosing $p(\cdot)$ to be a smooth non-decreasing function of $S_u$, parameters of which can be learnt by minimizing the loss function $\mathcal{L}(\kappa, p)$. We use the cumulative distribution function of an exponential distribution in our experiments, i.e, $p(X_u, S_u) = 1 - e^{-\beta \cdot S_u}$ (Figure 2a). The rate parameter, $\beta$, is learnt using gradient descent along with the weights of the neural network.

**Extension to $K$ Clusters**  Until now, we dealt with clustering the subjects into two groups. However, it is not hard to extend the framework for $K$ clusters. We increase the number of units in the output layer from 2 to $K$. As before, a softmax function applied at the output layer gives probabilities that define a soft clustering of the subjects into $K$ groups. These probabilities can be used to obtain the loss, $\mathcal{L}_{A,B}$, between any two groups, $\mathcal{D}_A$ and $\mathcal{D}_B$.

We define the total loss for $K$ groups as the average of all the pairwise losses between individual groups and get the geometric mean of the pairwise p-values, i.e.,

$$\mathcal{L}_{1\ldots K} = \frac{\sum_a \sum_{a \neq b} \mathcal{L}_{a,b}}{\binom{K}{2}} . \tag{10}$$

In other words, the loss $\mathcal{L}_{1\ldots K}$ is minimized only if each of the individual p-values are low indicating that each group's lifetime distribution is different (in divergence) from every other group's lifetime distribution.

**Implementation**  We implement a feedforward neural network in Theano (Theano Development Team, 2016) and use ADAM (Kingma and Ba, 2014) to optimize the loss $\mathcal{L}_{1\ldots K}$ defined in equation 10. Each iteration of the optimization takes as input a batch of subjects (full batch or a minibatch), generates a value for the loss, calculates the gradients, and updates the parameters $(\beta, \{W_i, b_i\}_{i=1}^L)$. This is done repeatedly until there is no improvement in the validation loss. We use L2 regularization over the weights and experiment with different values for the regularization parameter. We also experiment with different neural network sizes (number of hidden layers, number of hidden units), activation functions for the hidden layers, and weight initialization techniques. We applied different deep learning techniques like batch normalization (Ioffe and Szegedy, 2015) and dropout (Srivastava et al., 2014) to better learn the neural network.

## 5  RESULTS

**Dataset**  In this paper, we analyze a large-scale social network dataset collected from Friendster. After processing 30TB of data, originally collected by the Internet Archive in June 2011, the resulting network has around 15 million users with 335 million friendship links. Each user has profile information such as age, gender, marital status, occupation, and interests. Additionally, there are user comments on each other's profile pages with timestamps that indicate activity in the site.

In our experiments, we only use the data up to March 2008 as Friendster's monthly active users have been significantly affected with the introduction of "new Facebook wall" (Ribeiro and Faloutsos, 2015). From this, we only consider a subset of 1.1 million users who had participated in atleast one comment, and had specified their basic profile information like age and gender. We make our processed data available to the public at *location* (anonymized).

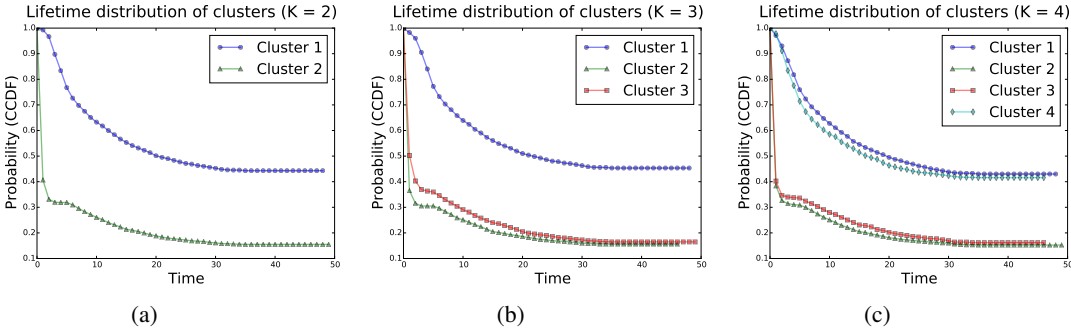

Figure 3: Empirical lifetime distributions (calculated using *end-of-life* signals) of clusters obtained from the proposed neural network approach for different values of $K$. The figures clearly indicate only two clusters in the Friendster social network : short-lived and long-lived users.

We build the dataset $\mathcal{D} : \{X_u, \{Y_{u,i}\}_{i=1}^{q_u}, S_u\}$ (Definition 1) for our clustering task as follows.
$X_u$ : We use each user's profile information (like age, gender, relationship status, occupation and location) as features. For the high-dimensional discrete attributes like location and occupation, we use 20 most frequently occurring values. To help improve the performance of competing methods, we also construct additional features using the user's activity over the initial 5 months (like number of comments sent and received, number of individuals interacted with, etc.). In total, we construct 60 features that are used for each of the models in our experiments.
$Y_{u,i}$ : We define $Y_{u,i}$ as the time between $u$'s $i$th comment (sent or received) and $(i-1)$st comment (sent or received). $q(u)$ is then defined as the total number of comments the user participated in.
$S_u$ : We calculate $S_u$ as the time between $u$'s last activity and the chosen time of measurement (March 2008).

**Model Training & Evaluation**   We experimented with different neural network architectures as shown in Table 2. In Table 1, we show the results for a simple neural network configuration with one fully-connected hidden layer with 128 hidden units and tanh activation function. We use a batch size of 8192 and a learning rate of $10^{-4}$. We also use batch normalization (Ioffe and Szegedy, 2015) to facilitate convergence, and regularize the weights of the neural network using an L2 penalty of 0.01. Appendix 8.2 shows a more detailed evaluation of different architecture choices.

We evaluate the models using 10-fold cross validation as follows. We split the dataset $\mathcal{D}$ randomly into 10 folds, sample 100,000 users without replacement from $i$th fold for testing and sample 100,000 users similarly from the remaining 9 folds for training. We use 20% of the training data as validation data for early stopping in our neural network training. We repeat this for $i$ ranging from 1 to 10.

We compare our clustering approach with the only two survival-based clustering methods in literature; a) **Semi-supervised clustering** (Bair and Tibshirani, 2004) and b) **Supervised sparse clustering** (Gaynor and Bair, 2013). Since both these methods require clear *end-of-life* signals, we use an arbitrary window of 10 months (i.e., a pre-defined "timeout") prior to the time of measurement in order to obtain these signals (Figure 2a). We also try window sizes of 0 months (only the users with activity at $t_m$ are censored) and 5 months, and obtain similar results (not reported here). We test our approach in two cases – in the presence and lack of *end-of-life* signals. In the former case, we optimize the loss function in equation (10) keeping $p(\cdot)$ fixed to the *end-of-life* signals obtained from using a window of 10 months. In the latter case, our approach learns the latent *end-of-life* signals. We also experiment with a loss function based on the Kolmogorov-Smirnov statistic (denoted NN-KS) and report the performance for the same. We evaluate the clusters obtained from each of these methods using concordance index.

**Concordance Index**   Concordance index or C-index (Harrell et al., 1982) is a commonly used metric in survival applications (Alaa and van der Schaar, 2017; Luck et al., 2017) to quantify a model's ability to discriminate between subjects with different lifetimes. It calculates the fraction of pairs of subjects for which the model predicts the correct order of survival while also incorporating censoring. We use the *end-of-life* signals calculated using a pre-defined "timeout" of 10 months. Rather than populating all possible pairs of users, we sample 10000 random pairs to calculate the C-index. Table 1 shows the C-index values for the baselines and the proposed method.

| Method | $K = 2$ | $K = 3$ | $K = 4$ |
|---|---|---|---|
| Semi Supervised Clustering (timeout) | $63.00 \pm 02.33$ | $64.68 \pm 08.52$ | $69.90 \pm 05.09$ |
| Supervised Sparse Clustering (timeout) | $67.71 \pm 02.69$ | $70.51 \pm 04.57$ | $68.83 \pm 05.99$ |
| Proposed NN-Kuiper (timeout) | $73.18 \pm 00.43$ | $73.61 \pm 01.91$ | $72.38 \pm 01.23$ |
| Proposed NN-Kuiper (learnt exponential) | $\mathbf{74.31 \pm 00.33}$ | $75.45 \pm 01.93$ | $73.64 \pm 00.92$ |
| Proposed NN-KS (learnt exponential) | $\mathbf{74.76 \pm 00.51}$ | $74.32 \pm 01.94$ | $72.83 \pm 00.75$ |

Table 1: C-index (%) for clusters from different methods with $K = 2, 3, 4$ where $K$ is the number of clusters. The proposed approach is more robust with lower values of standard deviations than the competing methods.

**Discussion**   The proposed neural network approach performs better on average than the two competing methods. Even without *end-of-life* signals, the proposed approach achieves comparable scores for $K = 3, 4$ and the best C-index score for $K = 2$. Although NN-Kuiper is theoretically more robust than NN-KS because of its increased statistical power in distinguishing distribution tails (Tygert, 2010), we do not observe a performance difference in the Friendster dataset. Further, we use the *end-of-life* signals obtained using a window of 10 months to calculate the empirical lifetime distributions of the clusters identified by the neural network (Figure 3). The empirical lifetime distributions of clusters seem distinct from each other at $K = 2$ but not at $K = 3, 4$. In addition, there is not significant gain in the C-index values as we increase the number of clusters from $K = 2$ to $K = 4$. Hence, we can conclude that there are only two types of users in the Friendster dataset – *long-lived* and *short-lived*.

## 6   RELATED WORK

Majority of the work in survival analysis has dealt with the task of predicting the survival outcome especially when the number of features is much higher than the number of subjects (Witten and Tibshirani, 2010b; Bøvelstad et al., 2007; Hothorn et al., 2006; Shivaswamy, Chu, and Jansche, 2007). A number of approaches have also been proposed to perform feature selection in survival data (Ishwaran et al., 2010; Lagani and Tsamardinos, 2010). In the social network scenario, Sun et al. (2012) tried to predict the relationship building time, that is, the time until a particular link is formed in the network.

Many unsupervised approaches have been proposed to identify cancer subtypes in gene expression data without considering the survival outcome (Eisen et al., 1998; Alizadeh et al., 2000; Bhattacharjee et al., 2001). Traditional semi-supervised clustering methods (Aggarwal, Gates, and Yu, 2004; Basu, Banerjee, and Mooney, 2002; Basu, Bilenko, and Mooney, 2004; Nigam et al., 1998) do not perform well in this scenario since they do not provide a way to handle the issues with right censoring. Semi-supervised clustering (Bair and Tibshirani, 2004) and supervised sparse clustering Witten and Tibshirani (2010a) use Cox scores (Cox, 1992) to identify features associated with survival. They treat these features differently in order to perform clustering based on the survival outcome.

Although there are quite a few works on using neural networks to predict the hazard rates of individuals (Eleuteri et al., 2003; 2007), to the best of our knowledge, there hasn't been a work on using neural networks for a survival-based clustering task. Recently, Alaa and van der Schaar (2017) proposed a nonparametric Bayesian approach for survival analysis in the case of more than one competing events (multiple diseases). They not only assume the presence of *end-of-life* signals but also the type of event that caused the *end-of-life*. Luck et al. (2017) optimize a loss based on Cox's partial likelihood along with a penalty using a deep neural network to predict the probability of survival at a point in time. Here we considered the task of clustering subjects into low-risk/high-risk groups without observing any *end-of-life* signals.

Extensive research has been done on what is known as frailty analysis, for predicting survival outcomes in the presence of clustered observations (Hougaard, 1995; Chuang et al., 2005; Huang and Wolfe, 2002). Although frailty models provide more flexibility in the presence of clustered observations, they do not provide a mechanism for obtaining the clusters themselves, which is our primary goal. In addition, our approach does not assume proportional hazards unlike most frailty models.

## 7 CONCLUSION

In this work we introduced a Kuiper-based nonparametric loss function, and a corresponding back-propagation procedure (which backpropagates the loss over clusters rather than the loss per training example). These procedures are then used to train a feedforward neural network to inductively assign observed subject covariates into $K$ survival-based clusters, from high-risk to low-risk subjects, without requiring an end-of-life signal. We showed that the resultant neural network produces clusters with better C-index values than other competing methods. We also presented the survival distributions of the clusters obtained from our procedure and concluded that there were only two groups of users in the Friendster dataset.

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

# 8 APPENDIX

## 8.1 PROOF OF THEOREM 1

We restate the theorem for convenience.

**Theorem 1** (Kuiper loss properties)**.** *From Definition 7, consider two clusters with true lifetime distributions $\hat{P}(U_1)$ and $\hat{P}(U_2)$. Assume an infinite number of samples/subjects. Then, the loss function defined in equation (7) has the following properties:*

(a) *If the two clusters have distinct lifetime distributions, i.e. $\hat{P}(U_1) \neq \hat{P}(U_2)$ then, either $\exists \hat{\kappa}, \hat{p}$ such that $\mathcal{L}(\hat{\kappa}, \hat{p}) \to -\infty$, or $\forall \kappa, p, \mathcal{L}(\kappa, p) \to 0$.*

(b) *If the two clusters have the same stochastic process $\Psi_u$ (Definition 1), $\Psi_u = \Psi_v$, for any two subjects $u$ and $v$, regardless of cluster assigments, then $\forall \kappa, p, \mathcal{L}(\kappa, p) \to 0$.*

Both parts (a) and (b) of our proof need definition 3 that translates the observed data $\mathcal{D}_u$ for subject $u$ into a stochastic process.

Proof of **(a)**: If the two clusters have distinct lifetime distributions, it means that the distributions of $T_0$ and $T_1$ in eq. (2) are different. Then, either the right-censoring $\delta$ in eq. (3) does not allow us to see the difference between $T_0$ and $T_1$, and then there is no mappings $\hat{p}$ and $\hat{\kappa}$ that can get the distribution of $S_0(t; \hat{\kappa}, \hat{p})$ and $S_1(t; \hat{\kappa}, \hat{p})$ to be distinct, implying an $\mathcal{L}(\kappa, p) \to 0$, as $n \to \infty$ as the observations come from the same distribution, making the Kuiper score asymptotically equal to one; **or** $\delta$ does allow us to see the difference and then, clearly $\hat{p} \equiv 0$ with a mapping $\hat{\kappa}$ that assigns more than half of the subjects to their correct clusters, which would allow us to see the difference in $H_0$ and $H_1$, would give Kuiper score asymptotically equal to zero. Thus, $\mathcal{L}(\kappa, p) \to -\infty$, as $n \to \infty$.

Proof of **(b)**: Because $\kappa$ only take the subject covariates as input, and there are no dependencies between the subject covariates and the subject lifetime in eq. (2), any clustering based on the covariates will be a random assignment of users into clusters. Moreover, from eq. (3), the censoring time of subject $u$, $S_u$, has the same distribution for both clusters because the RMPPs are the same. Thus, $H_0 \overset{d}{=} H_1$, i.e., $H_0$ and $H_1$ have the same distributions, and the Kuiper $p$-value test returns zero, $\mathcal{L}(\kappa, p) \to 0$, as $n \to \infty$.

## 8.2 Neural Network Architecture

Table 2 shows all the different parameters and the corresponding values used in our experiments. Concordance index scores of our approach over different values for number of hidden layers and number of hidden units is shown in Table 3, and that over different batch sizes and learning rates is shown in Table 4.

| Parameter | Values | Default |
|---|---|---|
| nHiddenLayers | [1, 2, 3] | 1 |
| nHiddenUnits | [128, 256, 512] | 128 |
| Minibatch Size | [256, 512, 1024, 2048, 8192] | 8192 |
| Learning Rate | $[10^{-4}, 10^{-2}, 10^{-1}, 1]$ | $10^{-4}$ |
| Activation | [tanh, relU] | tanh |
| Batch Normalization | [True, False] | True |
| L2 Regularization | $[10^{-2}, 10^{-1}]$ | $10^{-2}$ |
| Dropout | [0, 0.25, 0.5] | 0 |

Table 2: Different neural network architectures that we experiment with in our experiments. While varying the value of a set of parameters, we keep the other parameters fixed to the default value.

| | | nHiddenUnits | | |
|---|---|---|---|---|
| | | 128 | 256 | 512 |
| | 1 | $74.31 \pm 0.33$ | $74.85 \pm 0.52$ | $74.15 \pm 0.60$ |
| nHiddenLayers | 2 | $74.91 \pm 0.82$ | $74.76 \pm 1.10$ | $74.81 \pm 0.72$ |
| | 3 | $74.63 \pm 1.34$ | $74.43 \pm 0.55$ | $74.44 \pm 0.81$ |

Table 3: C-index (%) over different NN architectures for the proposed approach with Kuiper loss (learnt exponential) and K = 2.

| | | Batch Size | | | | |
|---|---|---|---|---|---|---|
| | | 256 | 512 | 1024 | 2048 | 8192 |
| | $10^{-4}$ | $72.93 \pm 6.07$ | $71.48 \pm 7.52$ | $70.93 \pm 7.86$ | $73.19 \pm 5.32$ | $74.31 \pm 0.33$ |
| Learning Rate | $10^{-2}$ | $57.23 \pm 1.36$ | $56.29 \pm 2.06$ | $57.60 \pm 1.96$ | $61.78 \pm 6.95$ | $72.99 \pm 4.88$ |
| | $10^{-1}$ | $59.41 \pm 4.48$ | $61.20 \pm 4.47$ | $63.80 \pm 3.66$ | $64.47 \pm 4.02$ | $68.73 \pm 3.65$ |
| | 1 | $61.37 \pm 3.82$ | $63.20 \pm 3.34$ | $61.86 \pm 4.31$ | $64.68 \pm 3.27$ | $67.10 \pm 4.28$ |

Table 4: C-index (%) over different learning rates and batch sizes for the proposed NN approach with Kuiper loss (with learnt exponential) and K = 2.

