# OpenReview forum: "A Deep Learning Approach for Survival Clustering without End-of-life Signals"
_ICLR.cc/2018/Conference — Reject_

### Official Review · AnonReviewer1 · 2017-11-27
**This manuscript presents a novel approach to survival clustering and compares it with some competing models.**

**Rating:** 6
**Confidence:** 1

**Review:**

Pros:
The paper is a nice read, clearly written, and its originality is well stated by the authors, “addressing the lifetime clustering problem without end-of-life signals for the first time”. I do not feel experienced enough in the field to evaluate the significance of this work.

The approach proposed in the manuscript is mainly based on a newly-designed nonparametric loss function using the Kuiper statistic and uses a feed-forward neural network to optimize the loss function. This approach does challenge some traditional assumptions, such as the presence of end-of-life signals or the artificial defined timeouts. Instead of giving a clear end-of-life signal, the authors specify a probability of end-of-life that permits us to take into account the associated uncertainty. By analyzing a large-scale social network dataset, it is shown that the proposed method performs better on average than the other two traditional models.

Cons:
I think that the main drawback of the paper is that the structure of the neural network and the deep learning techniques used for optimizing the loss function are not explained in sufficient detail.

---

> ### Author Response · Authors · 2018-01-05
> **Response to AnonReviewer1**
>
> We are glad you enjoyed the paper. We have added the following paragraph describing the neural network architecture and the deep learning methods we use. We have also reported results for the different design choices in Appendix.
>
> “We experimented with different neural network architectures as shown in Table 2. In Table 1, we show the results for a simple neural network configuration with one fully-connected hidden layer with 128 hidden units and tanh activation function. We use a batch size of 8192 and a learning rate of 10^{-4}. We also use batch normalization to facilitate convergence, and regularize the weights of the neural network using an L2 penalty of 0.01. Appendix 8.2 shows a more detailed evaluation of different architecture choices.”

---

### Official Review · AnonReviewer2 · 2017-11-29
**Brittle application of deep learning to statistical survival analysis**

**Rating:** 4
**Confidence:** 4

**Review:**

This paper discusses an application of survival analysis  in social networks.

While the application area seems to be pertinent, the statistics as presented in this paper are suboptimal at best. There is no useful statistical setup described (what is random? etc etc), the interplay between censoring and end-of-life is left rather fuzzy, and mentioned clustering approaches are extensively studied in the statistical literature in so-called frailty analysis. The setting is also covered in statistics in the extensive literature on repeated measurements  and even time-series analysis. It's up to the authors discuss similarities and differences of results of the present approach and those areas.

The numerical result is not assessing the different design decisions of the approach (why use a Kuyper loss?) in this empirical paper.

---

> ### Author Response · Authors · 2018-01-05
> **Response to AnonReviewer2**
>
> Thanks for your comments.
>
> 1.	We had described the underlying statistical setup in Appendix, essentially describing the activity times of a cluster of subjects using a Random Marked Point Process (RMPP). Following your feedback, we have moved it to a section in the paper called ‘Formal Framework’.
>
> 2.	(Frailty analysis) In our original draft, we followed prior work [Witten and Tibshirani, 2010; Gaynor and Bair, 2013] and refrained from comparing our approach with frailty models to avoid confusion w.r.t. the task at hand. But we now do see the benefit of clarifying the tasks, and thank the reviewer for asking us to do so. We added the following paragraph clarifying this difference.
>
> “Extensive research has been done on what is known as frailty analysis, for predicting survival outcomes in the presence of clustered observations. Although frailty models provide more flexibility in the presence of clustered observations, they do not provide a mechanism for obtaining the clusters themselves, which is our primary goal. In addition, our approach does not assume proportional hazards unlike most frailty models.”
>
> 3.	Censoring and ‘end-of-life’ are simply the two possibilities for each user. In the case where we have end-of-life signals, a subject could be “dead” or “censored” based on the signal. Similarly, when we do not have an end-of-life signal, there is a probability of the subject being “dead” or “censored” (in our case, we calculate this probability using S_u, the time till censoring).
>
> 4.	We have reported the results for different choices for the loss function - Kuiper loss vs Kolmogorov-Smirnov loss. Although, the difference in performance between the two loss functions is not significant in the Friendster dataset, Kuiper loss is theoretically better due to its increased statistical power in distinguishing distribution tails.
>
> 5.	We have also reported results for different neural network design choices (batch sizes, learning rates, number of hidden layers, and number of hidden units) in Appendix.

---

### Official Review · AnonReviewer3 · 2017-12-02
**A Deep Learning Approach for Survival Clustering without End-of-life Signals**

**Rating:** 6
**Confidence:** 5

**Review:**

Authors provide an interesting loss function approach for clustering using a deep neural network. They optimize Kuiper-based nonparametric loss and apply the approach on a large social network data-set.  However, the details of the deep learning approach are not well described. Some specific comments are given below.

1.Further details on use of 10-fold cross validation need to be discussed including over-fitting aspect.
2. Details on deep learning, number of hidden layers, number of hidden units, activation functions, weight adjustment details on each learning methods should be included.

3. Conclusion section is very brief and can be expanded by including a discussion on results comparison and  over fitting aspects in cross validation. Use of Kuiper-based nonparametric loss should also be justified as there are other loss functions can be used under these settings.

---

> ### Author Response · Authors · 2018-01-05
> **Response to AnonReviewer3**
>
> Thanks for your comments.
>
> 1.	We do not seem to be overfitting to the training data because: a) our loss function is not susceptible to outliers in the dataset (as it considers set distributions instead of the more standard approach of using a loss function defined over each individual data point), b) we monitor the validation loss while training the neural network, and c) we are able to generalize well in the test data.
>
> 2.	We added the following paragraph describing the deep learning techniques we used. Moreover, we now report results for different neural network design choices (batch sizes, learning rates, number of hidden layers, and number of hidden units).
>
> “We experimented with different neural network architectures as shown in Table 2. In Table 1, we show the results for a simple neural network configuration with one fully-connected hidden layer with 128 hidden units and tanh activation function. We use a batch size of 8192 and a learning rate of 10^{-4}. We also use batch normalization to facilitate convergence, and regularize the weights of the neural network using an L2 penalty of 0.01. Appendix 8.2 shows a more detailed evaluation of different architecture choices.”
>
> 3.	We now also report results for Kolmogorov-Smirnov loss. Although the difference in performance between the two loss functions is not significant in the Friendster dataset, Kuiper loss has higher statistical power in distinguishing distribution tails [Tygert 2010].

---

### Decision · Program_Chairs · 2018-01-29
**ICLR 2018 Conference Acceptance Decision**

**Decision:**

Reject

**Comment:**

The submission proposes a Kuiper statistic based loss function for survival clustering.  This loss function is applied to train a deep network.  Results are presented on a Friendster dataset.

This submission received borderline/mixed reviews.  The primary concerns were: justification of the Kuiper loss, lack of details of the experimental setup, writing style.  In the end, these concerns remain.

Of particular importance is the justification and experimental validation of the Kuiper statistic.  Although it seems a reasonable choice, from the authors' response to R3: "We now also report results for Kolmogorov-Smirnov loss. Although the difference in performance between the two loss functions is not significant in the Friendster dataset, Kuiper loss has higher statistical power in distinguishing distribution tails [Tygert 2010]."  If this theoretical result from [Tygert 2010] is relevant, it should be possible to demonstrate this experimentally.  If such differences are irrelevant for the data of interest, the paper should perhaps be reframed with a better discussion of available statistics and literature (cf. Reviewer 2), and a more general presentation de-emphasizing modeling choices that may have limited practical relevance.